# Excited-State Forces with GW-BSE Through the Hellmann–Feynman Theorem

**DOI:** 10.3390/ijms26052306

**Published:** 2025-03-05

**Authors:** Marah Jamil Alrahamneh, Iogann Tolbatov, Paolo Umari

**Affiliations:** 1Dipartimento di Fisica e Astronomia, Università di Padova, I-35131 Padova, Italy; 2CNR-IOM DEMOCRITOS, Istituto Officina dei Materiali, Consiglio Nazionale delle Ricerche, I-34149 Trieste, Italy

**Keywords:** density functional theory, photo-chemistry, many-body pertubation theory

## Abstract

We introduce a method for calculating the atomic forces of a molecular or extended system in an excited state described through the GW-BSE approach within the Tamm–Dancoff approximation. The derivative of the so-called excitonic Hamiltonian is obtained by finite differences and its application to the excited state is made possible through the use of suitable projectors. The scheme is implemented with the batch representation of the electron–hole amplitudes, allowing for avoiding sums over empty one-particle orbitals. The geometries of small excited molecules, namely, CO and CH_2_O, were in excellent agreement with the results from quantum chemistry methods.

## 1. Introduction

The so-called BSE method [1,2,3], which takes its name from the Bethe–Salpeter equation included in it, is nowadays a sort of golden standard for the calculation from first principles of neutral excitation and related properties. Probably, this is due, on the one hand, to its versatility, as it can be applied for both isolated and extended systems and, on the other hand, to its accuracy, which rivals that of quantum chemistry wavefunction methods [4]. With respect to the latter, the BSE method offers better scaling of the computational cost with respect to the system size [5,6]. The BSE method provides excitation energies defined as the difference between that of the excited state and that of the ground state. By adding the energy of the ground state, typically given by DFT, we can access the (total) energy of the excited state. The theory was originally formulated by treating the ions as fixed (clamped). Only more recently, in the context of solid-state physics, some methods were proposed [7,8,9,10] that include coupling with phonons, and hence treat atomic motions as harmonic.

The BSE method with fixed ions thus gives vertical excitation energies, which can be compared with absorption lines. In many systems, both molecular and bulk, the excited state is no longer in an equilibrium configuration with respect to the atomic positions. As a result, the system relaxes toward a new equilibrium position at a lower total energy, which can be compared with the photoluminescence lines. In some cases, this can even lead to the breaking of chemical bonds. For the prediction and characterization of such phenomena, it is, in principle, sufficient to calculate the forces acting on the ions when the system is in an excited state. These forces are simply defined as (minus) the derivatives of the excited-state energy (excitation energy plus ground-state energy) with respect to the atomic displacements.

In general, for a system with N atoms, such forces can be obtained using the finite difference approximation:(1)$$\frac{f\left(\ldots ,x_{i}+\Delta x_{i},\ldots \right)-f\left(\ldots ,x_{i}-\Delta x_{i},\ldots \right)}{2\Delta x}$$
where $x_{i}$ are the Cartesian coordinates of each atom. Each atom is displaced by a small fixed distance ±Δx along the three Cartesian directions, resulting in 6N BSE energy calculations. This becomes computationally prohibitive, even for small *N*, given the significant cost of the BSE method compared with DFT. It is worth noting that in the case of systems displaying geometrical symmetry, we can reformulate the calculation in terms of a smaller number of internal generalized coordinates.

In principle, the forces acting on the atoms can also be calculated using the Hellman–Feynman theorem. These are just the expectation values of the derivatives of the so-called excitonic Hamiltonian with respect to the atomic displacement changed in sign. The expectation values are calculated for the excitonic states described by the BSE amplitudes. The first implementation of this approach was carried out by Ismail-Beigi in Ref. [11]. This requires the analytical derivation of the different terms of the BSE Hamiltonian, which, on the one hand, requires significant implementation effort and, on the other hand, is still slow because, again, 3N derivatives need to be computed. Ismail-Beigi’s work makes use of density functional perturbation theory (DFPT) [12] to evaluate the derivative of the excitonic Hamiltonian. In this work, we show how we can derive the BSE Hamiltonian using finite differences. This requires the application of the BSE Hamiltonian relative to a system with a displaced atom to an excited state that corresponds to the atoms in their original position. Since the excited states in the BSE method are treated with the Tamm–Dancoff approximation, i.e., written as a sum of Slater determinants relative to single valence–conduction promotions, only a subset of all the possible many-body excited states can be represented for a given atomic configuration. Thus, we identified an appropriate projector to apply the Hamiltonian corresponding to the displaced system. We implemented this in a scheme that does not require explicit summations over conduction states [6,13,14,15]. The implementation requires much less effort compared with the method of Ref. [11], as it is limited to the projectors. This simplicity makes it easier to incorporate it into various BSE codes. Compared with finite differences, the use of this Hellman–Feynman scheme requires only one BSE calculation. This becomes particularly advantageous when calculating the forces for many excited states, for example, to account for level crossing in complex systems.

## 2. Results

We implemented our approach in the GWL code [6,13,16,17,18], which is part of the Quantum Espresso package (see Refs. [19,20]). This is based on the planewave pseudo-potentials paradigm. The code allows the GW and BSE calculations to avoid any explicit sum over empty KS orbitals. This implies converged calculations, even for large model structures.

### 2.1. Carbon Monoxide

As a simple first test case, we considered an isolated CO molecule. This choice benefited from a large number of published results from accurate quantum chemistry calculations. The starting DFT run was performed using the so-called PBE approximation [21] for the exchange and correlation functional and norm-conserving pseudo-potentials. The molecule was placed in a cubic simulation cell with a 20 Bohr edge. A 70 Ry energy cutoff defined the planewaves basis set used for representing the wavefunctions. The irreducible polarizability operator was represented using an *optimal* basis set that comprised 50 elements [16]. GW quasi-particle energies were computed for the five occupied states and the five lowest unoccupied states. These energies, together with a rigid energy shift for the rest of the unoccupied states, were used for the diagonal term of Equation (Equation 9). We calculated the lowest singlet excited states at the BSE level. For the correct labeling of the BSE excited states, we devised and implemented an analysis tool that will be presented elsewhere. As it is customary, we used the letters Σ, Π, and Δ to denote the eigenstates of ${\hat{L}}_{x}$ with eigenvalues 0, 1, and 2, provided that *x* is the internuclear axis. We then labeled with + or − the Σ states that did not (did) change sign upon reflection through a plane containing the internuclear axis. The additional label used for ordering the states was taken from the literature as in Ref. [22].

In Figure 1, we display the total energy of the CO molecule in its ground state ($X^{1}\Sigma^{+}$) and in the lowest singlet states ($A^{1}\Pi$, $I^{1}\Sigma^{-}$, $D^{1}\Delta$). While the Σ states were not degenerate, all the others were doubly degenerate.

The corresponding vertical transition energies relative to the ground-state internuclear distance are reported in Table 1, together with previous theoretical calculations and experimental values. Our GW-BSE figures are in line with a previous study [23]. Both GW-BSE results overestimated the separation between the $A^{1}\Pi$ and $I^{1}\Sigma^{-}$ states. In contrast, this is well reproduced by advanced quantum chemistry calculations [22]. It is worth noting that the symmetry of the excited states is not explicitly indicated in Ref. [23].

We calculated the atomic forces using our new approach shown in Equation (Equation 16), where $\Delta_{\lambda }=0.1$ Bohr was used for evaluating the derivative of the excitonic Hamiltonian. We also used the same displacement for calculating atomic forces by *brute force* as finite differences of excited state energies. Taking advantage of the symmetry, we set the C-O axis along the *x* one and considered only displacements along that axis. The results are reported in Figure 2. We see that for internuclear C-O distances from 2.2 to 2.6 Bohr, the two approaches yielded results within 0.01 Ry/Bohr, validating our method. Interestingly, at shorter C-O distances, the simple differentiation through finite differences of exciton energies became cumbersome (disks, dotted lines). This was due the large degree of level crossing between the various excited state energy levels (see Appendix A). This required the evaluation of the overlaps between the excitons and their arrangement in the correct order. It is worth noting that our approach does not suffer from such a limitation, as the derivative of the excitonic Hamiltonian in Equation (Equation 16) is always well defined. We implemented a module for optimizing the structural configuration of a molecule in a given excited state. The total excited state energy was minimized using either the steepest descent or the Broyden—Fletcher—Goldfarb—Shanno (BFGS) algorithm [25].

We could identify the optimized position when the atomic forces are smaller than $10^{-2}$ Ry/Bohr. In Table 2, we report the equilibrium bond length for the ground state and the lowest excited states. The ground state $X^{1}\Sigma^{+}$ and the first singlet states were within 3% from the experimental values. This means that we retained the typical accuracy of DFT with semi-local exchange and correlation potentials. Although GW-BSE cannot attain the accuracy of multi-reference quantum chemistry approaches [22,26], it brings DFT accuracy to the realm of excitations.

### 2.2. Formaldehyde

To further assess our approach, we considered the formaldehyde molecule (CH_2_O), a commonly used test molecule that serves as a model for carbonyl dyes due to its distinct low-energy $n\to \pi^{*}$ excitation, which is well separated from higher-lying excited states [27]. Moreover, unlike the CO molecule, which possesses only one degree of freedom, formaldehyde is characterized by four independent parameters: the CO and CH bond lengths; the HCH angle; and the dihedral angle, which represents the deviation of the C-O bond from the $CH_{2}$ plane. Upon photoexcitation, the molecule displays a change in symmetry from the planar $C_{2v}$ group to the spatial $C_{s}$ one. We proceeded as for the CO molecule: we adopted, at the DFT level, the PBE approximation, with a cutoff of 70 Ry to define the planewave basis set and an edge of 20 Bohr for the periodic cubic simulation cell. The optimal basis, which was used for representing the polarizability matrices, comprised 50 elements and we calculated the quasi-particle energies for the 6 valence and lowest 15 conduction orbitals.

Our computations yielded a vertical excitation energy of 2.93 eV. The not negligible discrepancy with respect to the reference theoretical value of 3.89 eV, from highly accurate multi-reference quantum chemical calculations [28], was traced back to the lack of self-consistency in the GW step. Indeed, similar GW-BSE approaches yielded 2.83 eV in Ref. [23], 3.12 eV in Ref. [28], and 3.15 eV in Ref. [29].

We then optimized the geometrical configuration of the first excited state. The results are reported in Table 3. The bond lengths and the H-C-H angle were within less than 1% from the reference results from correlated quantum chemistry approaches (CC3) and from variational quantum Monte Carlo (VMC). Larger deviations of the order of 10% were found for the dihedral (θ) angle. This was due to the weak energy dependence with respect to dihedral angle variations. Indeed, two-dimensional scans of the excited-state potential energy surface (see Figure S2) confirmed that the energy was relatively flat along the dihedral direction. This explains why both our result and other high-level methods (CC3, VMC) show notable yet still energetically insignificant differences in θ.

## 3. Method

In the following, we rely on the Tamm–Dancoff approximation. The starting point is the one-particle Kohn–Sham (KS) orbitals:(2)$${\hat{H}}_{KS}\left|\phi_{i}\right.\rangle =\epsilon_{i}\left|\phi_{i}\right.\rangle$$
where $\phi_{i}$ is the *i*-th eigenstate of the Kohn–Sham Hamiltonian ${\hat{H}}_{KS}$. We consider the many-body ground state as the single Slater’s determinant formed by the occupied KS orbitals:(3)$$\left|\Psi_{0}\right.\rangle =\frac{1}{\sqrt{N_{v}!}}\prod_{v=1,N_{v}}{\hat{a}}_{v}^{†}\left|0\right.\rangle$$
where *v* runs over the $N_{v}$ valence states. Within the Tamm–Dancoff approximation, a generic excited state Ψ is written as a sum of Slater’s determinants with an occupied valence orbital substituted with an unoccupied conduction one:(4)$$\left|\Psi_{exc}\right.\rangle =\sum_{v,c}A_{v,c}{\hat{a}}_{c}^{†}{\hat{a}}_{v}\left|\Psi_{0}\right.\rangle$$
where *c* runs over the unoccupied orbitals. In principle, in Equation (Equation 4), the sum should be extended over all the conduction orbitals. As this becomes highly impractical for large systems, often only the lowest-energy empty orbitals are included. However, this can hinder the convergence. This issue can be overcome using approaches based on the so-called resolution of the identity operator [6,13,14,16]. Indeed, all the $A_{v,c}$ terms for a given *v* can be derived by a single wavefunction $\xi_{v}$, usually referred to as a *batch*, which belongs to the manifold of the empty states:(5)$$A_{v,c}=\langle \phi_{c}|\xi_{v}\rangle$$

To indicate the excited states $\Psi_{exc}$, we can introduce the following notation:(6)$$|\Psi_{exc}\rangle =|\left\{\xi_{v}\right\}\rangle$$

Then, the entire formalism is structured in such a way that all sums over empty orbitals are replaced by applications of the KS Hamiltonian to opportune wavefunctions.

Finally, we note that a generic many-body state can be written approximately as the ground state plus an excited state given by the Tamm–Dancoff approximation:(7)$$\left|\Psi \right.\rangle =\alpha \left|\Psi_{0}\right.\rangle +\left|\Psi_{exc}\right.\rangle$$
with $\left|\Psi_{exc}\right.\rangle$ not normalized.

The BSE approach [1,2,3] permits defining a so-called excitonic Hamiltonian ${\hat{H}}^{exc}$, which is composed of three terms:(8)$${\hat{H}}^{exc}=\hat{D}+{\hat{H}}^{x}+{\hat{H}}^{d}$$

The action of the diagonal term $\hat{D}$ on an excitation $|\left\{\xi_{v}\right\}\rangle$ reads [6](9)$$\begin{array}{cc} \hat{D}|\left\{\xi_{v}\right\}\rangle & =\left|\{\xi_{v}^{\prime }\right\}\rangle \\ |\xi_{v}^{\prime }\rangle & =\left({\hat{H}}_{KS}-\epsilon_{v}\hat{I}+\Delta \hat{I}\right)|\xi_{v}\rangle \end{array}$$
with $\hat{I}$ as the identity operator and Δ as a scissor energy in order to open the KS HOMO–LUMO gap to a better estimate usually given by a GW or another post-DFT approach [13]. Here and in the following, we consider real orbitals for the sake of simplicity. The action of the exchange term ${\hat{K}}^{x}$ reads(10)$$\begin{array}{cc} {\hat{K}}^{x}|\left\{\xi_{v}\right\}\rangle & =\left|\{\xi_{v}^{\prime \prime }\}\right.\rangle \\ \xi_{v}^{\prime \prime }\left(r\right) & =\int dr^{\prime }P_{c}\left(r,r^{\prime }\right)\phi_{v}\left(r^{\prime }\right)\sum_{v^{\prime }}\left[\int dr^{\prime \prime }v\left(r^{\prime },r^{\prime \prime }\right)\phi_{v^{\prime }}\left(r^{\prime \prime }\right)\xi_{v^{\prime }}\left(r^{\prime \prime }\right)\right] \end{array}$$
with ${\hat{P}}_{c}=\hat{I}-\sum_{v}|\phi_{v}\rangle \langle \phi_{v}|$ as the projector over the empty KS manifold and $\hat{v}$ as the bare Coulomb potential. The direct term ${\hat{K}}^{d}$ reads(11)$$\begin{array}{cc} {\hat{K}}^{d}|\left\{\xi_{v}\right\}\rangle & =\left|\{\xi_{v}^{\prime \prime \prime }\}\right.\rangle \\ \xi_{v}^{\prime \prime \prime }\left(r\right) & =-\int dr^{\prime }P_{c}\left(r,r^{\prime }\right)\sum_{v^{\prime }}\xi_{v^{\prime }}\left(r^{\prime }\right)\left[\int dr^{\prime \prime }W\left(r^{\prime },r^{\prime \prime }\right)\phi_{v^{\prime }}\left(r^{\prime \prime }\right)\phi_{v}\left(r^{\prime \prime }\right)\right] \end{array}$$
where $\hat{W}$ is the static (i.e., at zero frequency) screened Coulomb interaction. The excitonic wavefunctions are found by solving the excitonic Hamiltonian through diagonalization:(12)$$\left.{\hat{H}}^{exc}\right|\Psi_{S}\rangle =\Omega_{S}|\Psi_{S}\rangle$$
with S=1,2,… as an index running over the excitonic eigenfunction and $\Omega_{S}$ as the corresponding excitation energy. This means that the energy $E_{S}$ of the *S*-th excited state yields(13)$$E_{S}=E_{0}+\Omega_{S}$$
with $E_{0}$ as the ground-state energy given by DFT.

We suppose now that the Hamiltonian operator of our system depends on an external parameter λ with the unperturbed configuration given by λ=0. This parameter could be the displacement of an atom from its equilibrium position or the magnitude of an electric field coupled with the system. The ground- $E_{0}\left(\lambda \right)$ and excited-state energies $E_{S}\left(\lambda \right)$ are now functions of λ as the excitation energy $\Omega_{S}\left(\lambda \right)$. We are interested in the response of the (excited) system at the linear order:(14)$$\frac{\partial E_{S}}{\partial \lambda }=\frac{\partial E_{0}}{\partial \lambda }+\frac{\partial \Omega_{S}}{\partial \lambda }$$

While the derivative of the ground-state energy is evaluated directly from DFT, the derivative of the excitation energy can be obtained from the Hellman–Feynman theorem [11]:(15)$$\frac{\partial \Omega_{S}}{\partial \lambda }=\langle \Psi_{S}\left|\frac{\partial {\hat{H}}^{exc}}{\partial \lambda }\right|\Psi_{S}\rangle$$

In contrast with Ref. [11], we calculate the derivative of the excitonic Hamiltonian using differences:(16)$$\frac{\partial {\hat{H}}^{exc}}{\partial \lambda }≃\frac{{\hat{H}}^{exc}\left(\Delta_{\lambda }\right)-{\hat{H}}^{exc}\left(-\Delta_{\lambda }\right)}{2\Delta_{\lambda }}$$
where we consider finite displacements $\pm \Delta_{\lambda }$ over λ, and the state $\Psi_{S}$ is an eigenstate of ${\hat{H}}^{exc}\left(\lambda =0\right)$. It is important to note that the manifold of the Tamm–Dancoff (excited) states depends on λ. If we want to use Equation (Equation 15), we must first define a method for projecting a state relative to λ=0 on the manifold for an arbitrary λ. As the KS Hamiltonian is also a function of λ, we write(17)$${\hat{H}}_{KS}\left(\lambda \right)\left|\phi_{i}\left(\lambda \right)\right.\rangle =\epsilon_{v}\left(\lambda \right)\left|\phi_{i}\left(\lambda \right)\right.\rangle$$

We define $\hat{P}\left(\lambda \right)$ as the operator that projects excited states on the manifold for λ=0 onto the manifold for λ:(18)$$\hat{P}\left(\lambda \right)|\Psi_{exc}\rangle =\sum_{v,c}{\hat{a}}_{c}^{†}\left(\lambda \right){\hat{a}}_{v}\left(\lambda \right)\left|\Psi_{0}\left(\lambda \right)\right.\rangle \langle \Psi_{0}\left(\lambda \right)|{\hat{a}}_{v}^{†}\left(\lambda \right){\hat{a}}_{c}\left(\lambda \right)|\Psi_{exc}\rangle$$
where $\Psi_{0}\left(\lambda \right)$ is the many-body ground state for a given value of λ, and ${\hat{a}}_{c,v}\left(\lambda \right)$ and ${\hat{a}}_{c,v}^{†}\left(\lambda \right)$ are the corresponding annihilator and creator operators. It is easy to show that the corresponding (unnormalized) coefficients ${\tilde{A}}_{v^{\prime },c^{\prime }}\left(\lambda \right)$ read as(19)$${\tilde{A}}_{v^{\prime },c^{\prime }}\left(\lambda \right)=\sum_{v,c}A_{v,c}\det\left|\langle \phi_{1}\left(\lambda \right)\ldots \phi_{v^{\prime }-1}\left(\lambda \right)\phi_{c}^{\prime }\left(\lambda \right)\phi_{v^{\prime }+1}\left(\lambda \right)\ldots \phi_{N_{v}}\left(\lambda \right)|\phi_{1}\ldots \phi_{v-1}\phi_{c}\phi_{v+1}\ldots \phi_{N_{v}}\rangle \right|$$

Now, we want to rewrite the coefficients ${\tilde{A}}_{v^{\prime },c^{\prime }}\left(\lambda \right)$ in terms of *batches* in order to avoid explicit sums or indices over the empty orbitals. First, we write(20)$${\tilde{A}}_{v^{\prime },c^{\prime }}\left(\lambda \right)=\sum_{v}\det\left|\langle \phi_{1}\left(\lambda \right)\ldots \phi_{v^{\prime }-1}\left(\lambda \right)\phi_{c^{\prime }}\left(\lambda \right)\phi_{v^{\prime }+1}\left(\lambda \right)\ldots \phi_{N_{v}}\left(\lambda \right)|\phi_{1}\ldots \phi_{v-1}\xi_{v}\phi_{v+1}\ldots \phi_{N_{v}}\rangle \right|$$

Now, we want to find the corresponding (unnormalized) batches $\tilde{\xi }\left(\lambda \right)$ relative to the manifold at a given λ. With some algebra, we can write(21)$$\begin{array}{cc} |{\tilde{\xi }}_{v^{\prime }}\left(\lambda \right)\rangle & =\sum_{v}{\left(-1\right)}^{v^{\prime }-1}\times (\\ \; & \sum_{v^{\prime \prime }<v}{\left(-1\right)}^{v-1}{\left(-1\right)}^{v^{\prime \prime }}{\hat{P}}_{c}\left(\lambda \right)|\phi_{v^{\prime \prime }}\rangle \det|\langle \phi_{1}\left(\lambda \right)\phi_{v^{\prime }-1}\left(\lambda \right)\phi_{v^{\prime }+1}\left(\lambda \right)\ldots \phi_{N_{v}}\left(\lambda \right)|\xi_{v}\phi_{1}\ldots \phi_{v^{\prime \prime }}\ldots \phi_{v-1}\phi_{v+1}\ldots \phi_{N_{v}}\rangle |+\\ \; & -\sum_{v^{\prime \prime }>v}{\left(-1\right)}^{v-1}{\left(-1\right)}^{v^{\prime \prime }}{\hat{P}}_{c}\left(\lambda \right)|\phi_{v^{\prime \prime }}\rangle \det|\langle \phi_{1}\left(\lambda \right)\phi_{v^{\prime }-1}\left(\lambda \right)\phi_{v^{\prime }+1}\left(\lambda \right)\ldots \phi_{N_{v}}\left(\lambda \right)|\xi_{v}\phi_{1}\ldots \phi_{v^{\prime \prime }}\ldots \phi_{v-1}\phi_{v+1}\ldots \phi_{N_{v}}\rangle |+\\ \; & {\left(-1\right)}^{v-1}{\hat{P}}_{c}\left(\lambda \right)|\xi_{v}\rangle \det|\langle \phi_{1}\left(\lambda \right)\phi_{v^{\prime }-1}\left(\lambda \right)\phi_{v^{\prime }+1}\left(\lambda \right)\ldots \phi_{N_{v}}\left(\lambda \right)|\phi_{1}\ldots \phi_{v-1}\phi_{v+1}\ldots \phi_{N_{v}}\rangle \left|\right) \end{array}$$
where we introduce the projector over the KS manifold of empty orbitals ${\hat{P}}_{c}\left(\lambda \right)$:(22)$${\hat{P}}_{c}\left(\lambda \right)=1-\sum_{v^{\prime }=1,N_{v}}|\phi_{v^{\prime }}\left(\lambda \right)\rangle \langle \phi_{v^{\prime }}\left(\lambda \right)|$$

Interestingly, the overlap between the excited state $|\Psi_{S}\rangle$ and the ground state $|\Psi_{0}\left(\lambda \right)$ is in general not zero for a finite λ. We call g(λ) such an overlap:(23)$$\begin{array}{cc} g(\lambda ) & =\langle \Psi_{0}\left(\lambda \right)|\Psi_{S}\rangle \\ \; & =\sum_{v}\det\langle \phi_{1}\left(\lambda \right)\ldots \phi_{N_{v}}\left(\lambda \right)|\phi_{1}\ldots \phi_{v-1}\xi_{v}\phi_{v+1}\ldots \phi_{N_{v}}\rangle \end{array}$$

Finally, the derivative of the excited-state energy yields(24)$$\begin{array}{cc} \frac{\partial E_{S}\left(\lambda \right)}{\partial \lambda } & ≃\frac{1}{2\Delta_{\lambda }}\left(\langle \Psi_{S}|{\hat{P}}^{†}\left(\Delta_{\lambda }\right){\hat{H}}_{exc}\left(\Delta_{\lambda }\right)\hat{P}\left(\Delta_{\lambda }\right)|\Psi_{S}\rangle \right.\\ \; & -\left.\langle \Psi_{S}|{\hat{P}}^{†}\left(-\Delta_{\lambda }\right){\hat{H}}_{exc}\left(-\Delta_{\lambda }\right)\hat{P}\left(-\Delta_{\lambda }\right)|\Psi_{S}\rangle \right)\\ \; & +\frac{\partial E_{0}\left(\lambda \right)}{\partial \lambda } \end{array}$$
where we make use of the property g(Δλ)=−g(−Δλ), which holds in the linear regime.

## 4. Conclusions

We proved how our method for the projection of excitons can be successfully used for computing atomic forces of molecular systems in an excited state. This can be particularly advantageous for the cases where taking finite differences of total energies becomes cumbersome. This happens, for example, in a large system as a consequence of level crossing. The latter is completely avoided by our approach, which involves finite differences of the excitonic Hamiltonian.

Another straightforward application would be the evaluation of electric dipoles [32]. These are defined as the derivative of the total energy of a system with respect to the internal macroscopic electric field so that we can take finite differences of the excitonic Hamiltonian with respect to the electric field.

A merit of our approach is its simplicity so that it would be straightforward to implement it in other BSE codes. Finally, the algorithm for projecting excitons would permit real-time time-dependent GW-BSE simulations [33] in the presence of a time-varying potential. In turn, this could be applied to schemes such as surface hopping [34] to reach much longer timescales.

## Figures and Tables

**Figure 1 ijms-26-02306-f001:**
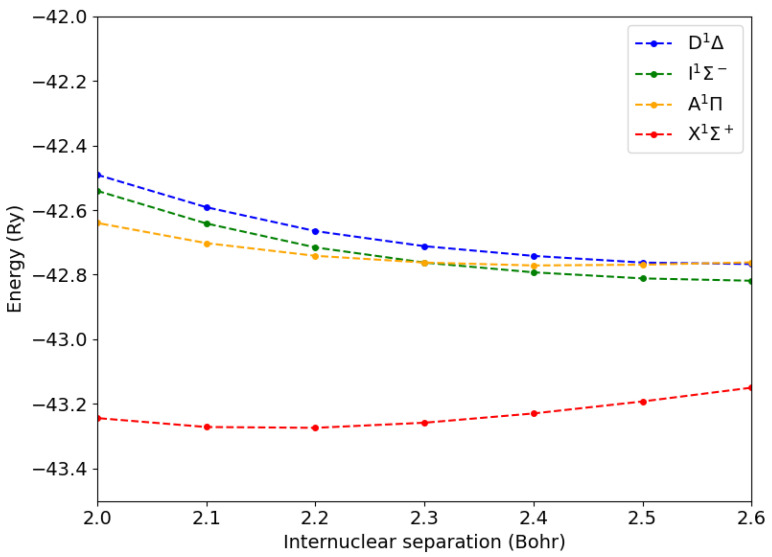
Total energies of the CO molecule in the ground state ($X^{1}\Sigma^{+}$) and in the lowest excited states as a function of the C-O internuclear distance.

**Figure 2 ijms-26-02306-f002:**
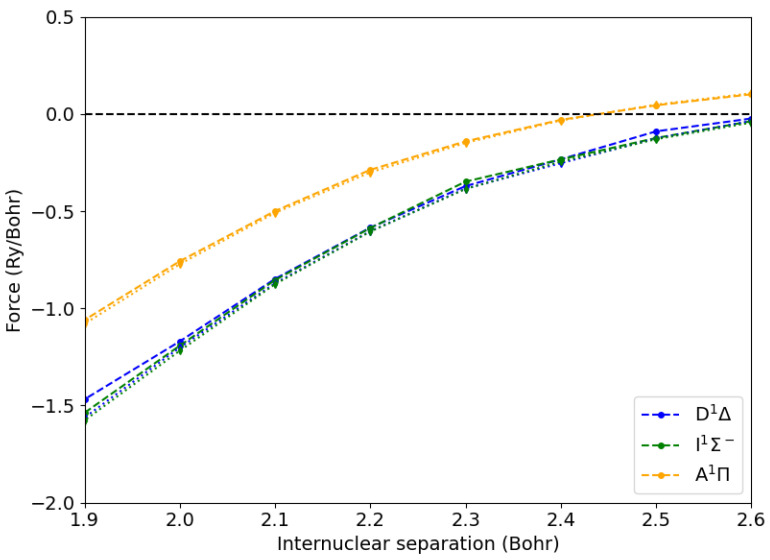
Atomic forces from Hellmann–Feynman (disks, dashed lines) and numerical differentiation (diamonds, dotted lines) methods for the carbon atom of a CO molecule in the ground state ($X^{1}\Sigma^{+}$) and in the lowest excited states as a function of the C-O internuclear distance.

**Table 1 ijms-26-02306-t001:** Vertical transition energies for the CO molecule with respect to the ground state. Figures are in eV.

Method	$A^{1}\Pi$	$I^{1}\Sigma^{-}$	$D^{1}\Delta$
This	7.43	7.95	8.65
GW-BSE [23]	7.67	8.24	
CASSCF-icMRCI [22]	8.14	8.19	8.48
Exp. [24]	8.07	8.07	8.17

**Table 2 ijms-26-02306-t002:** C-O internuclear distance at equilibrium for the ground state ($X^{1}\Sigma^{+}$) and the first excited states $A^{1}\Pi$, $I^{1}\Sigma^{-}$, and $D^{1}\Delta$ of the CO molecule. Figures are in Bohr.

Method	$X^{1}\Sigma^{+}$	$A^{1}\Pi$	$I^{1}\Sigma^{-}$	$D^{1}\Delta$
This	2.16	2.43	2.60	2.60
MR-CISD+Q [26]	2.14	2.34	2.65	
CASSCF-icMRCI [22]	2.14	2.34	2.61	2.65
Exp. [24]	2.13	2.33	2.63	2.64

**Table 3 ijms-26-02306-t003:** Geometrical configuration of the CH_2_O molecule in the first excited state; bond-lengths are given in Bohr and angles in degrees.

Method	$R_{\mathrm{CH}}$	$R_{\mathrm{CO}}$	Angle HCH	Angle θ
This	2.04	2.50	117.3	30.8
CC3 [30]	2.06	2.51	118.3	36.8
VMC [31]	2.05	2.52	119.6	33.3

## Data Availability

The original contributions presented in this study are included in the article/Appendix A. Further inquiries can be directed to the corresponding author.

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
