# Peer review of "Excited-State Forces with GW-BSE Through the Hellmann–Feynman Theorem"

_ijms, 2025, doi:10.3390/ijms26052306_

Round 1
Reviewer 1 Report
Comments and Suggestions for Authors
Please see the attached file.

Reviewer 2 Report
Comments and Suggestions for Authors
The paper by Alrahamneh and co-workers presents a new method based to calculate atomic forces at the GW-BSE level. The new method is easier to implement compared to the earlier methods. It shows good agreement with either the numerical forces and other theories in the literature on CO and formaldehyde. I recommend publication of this paper after the authors consider improving the paper by the following comments:
1. In line 46, the authors mention that 6N calculations are required to calculate the forces by finite difference. Do the authors refer to (f(x1+Δx, x2, …)-f(x1-Δx, x2, …))/2Δx, where xi are generalized coordinates and Δx is the displacement? If so, it would be a good idea to clarify this in the introduction, since similar ideas are not mentioned until eq 18. In addition, a molecule has only 3N-5 or 3N-6 degrees of freedom, since some of the 3N degrees of freedom are redundant. In principle one could calculate the forces in internal coordinates, then transformed into Cartesian. For example, one would do two calculations for a diatomic molecule over the internuclear distance, rather than 12 calculations over the Cartesian coordinates. One could do 10 or 12 less calculations than the 6N approach.
2. Following the comments in 1, since the author’s new method also requires finite difference calculations, I wonder if the CO force calculations are done in Cartesian or internal coordinates (internuclear distance)? In other words, are 2 calculations performed for each internuclear distance, or does it require 12 calculations?
2. In line 56, the authors mention that the Ismail-Beig’s method in ref 11 requires 6N derivative calculations. I wonder if it is a typo, which should be 3N, because the forces are 3N dimensional vector, which corresponds to x, y, z directions for each atom.
3. The main advantage of the new method compared to Ismail-Beig’s method is that it requires less implementation effort. Since the authors mentioned Ismail-Beig’s method is still computationally expensive (line 55), I wonder if the new method has any advantage on the computational cost and/or accuracy compared to Ismail-Beig’s method? It would be interesting to compare the new approach with Ismail-Beig’s method side by side, if possible.
4. The authors should be clear about which level of theory/experiment are used to obtain the potential energy curves in Fig 1 and A1. Are they BSE calculations by the authors using PBE or is it obtained from a reference?
5. In the example calculation on formaldehyde, the authors only compare to other methods in the literature. While the results do agree well, it does not rigorously show the accuracy of the new force calculation method, which should be the most important point of the discussion. One needs to rule out the error of the underlying electronic structure theory. It would be helpful to perform a numerical force calculation at the optimized structure and see if the numerical forces are reasonably close to zero within the convergence threshold.
6. The reference in line 203 should be fixed.
